# Aberrant integration of Hepatitis B virus DNA promotes major restructuring of human hepatocellular carcinoma genome architecture

Eva G. Álvarez [1,2], Jonas Demeulemeester [3,4,23], Paula Otero [1,2,23], Clemency Jolly[3,23],
Daniel García-Souto [1,2,23], Ana Pequeño-Valtierra[1], Jorge Zamora[1], Marta Tojo[5], Javier Temes [1],
Adrian Baez-Ortega[6], Bernardo Rodriguez-Martin [1,2], Ana Oitaben [1,2], Alicia L. Bruzos [1,2],
Mónica Martínez-Fernández[1], Kerstin Haase [3], Sonia Zumalave [1,2], Rosanna Abal[1],
Jorge Rodríguez-Castro [1], Aitor Rodriguez-Casanova [7,8], Angel Diaz-Lagares[7,9], Yilong Li[10],
Keiran M. Raine [10], Adam P. Butler[10], Iago Otero [1,2], Atsushi Ono[11], Hiroshi Aikata[11], Kazuaki Chayama[12,13,14],
Masaki Ueno[15], Shinya Hayami[15], Hiroki Yamaue[15], Kazuhiro Maejima[14], Miguel G. Blanco [1], Xavier Forns[16],
Carmen Rivas[1,17], Juan Ruiz-Bañobre [1,9,18,19], Sofía Pérez-del-Pulgar [16], Raúl Torres-Ruiz [20,21],
Sandra Rodriguez-Perales [20], Urtzi Garaigorta [17,24], Peter J. Campbell [10,22,24], Hidewaki Nakagawa[14,24],
Peter Van Loo [3,24] & Jose M. C. Tubio [1,2✉]

Most cancers are characterized by the somatic acquisition of genomic rearrangements during tumour evolution that eventually drive the oncogenesis. Here, using multiplatform sequencing technologies, we identify and characterize a remarkable mutational mechanism in human hepatocellular carcinoma caused by Hepatitis B virus, by which DNA molecules from the virus are inserted into the tumour genome causing dramatic changes in its configuration, including non-homologous chromosomal fusions, dicentric chromosomes and megabase-size telomeric deletions. This aberrant mutational mechanism, present in at least 8% of all HCC tumours, can provide the driver rearrangements that a cancer clone requires to survive and grow, including loss of relevant tumour suppressor genes. Most of these events are clonal and occur early during liver cancer evolution. Real-time timing estimation reveals some HBV-mediated rearrangements occur as early as two decades before cancer diagnosis. Overall, these data underscore the importance of characterising liver cancer genomes for patterns of HBV integration.

A full list of author affiliations appears at the end of the paper.

Human hepatocellular carcinoma (HCC) is the most common primary liver malignancy, resulting in over 700,000 deaths globally every year[1]. Previous studies indicate that the disease has a complex genomic landscape, with frequent copy number changes and interchromosomal rearrangements[2,3]. Hepatitis B virus (HBV) infection – a condition affecting 240 million people worldwide – is the second most frequent cause of cancer after tobacco, and a major cause of HCC. HBV infection has been associated with chromosomal instability in cancerous and non-cancerous liver genomes, and HBV DNA integration is known to be the cause of chromosomal rearrangements in HCC[4–11]. However, we still ignore the full extent to which HBV DNA integrations impact the structure (i.e., patterns and mechanisms of mutation) and function (i.e., driver events) of HCC genomes[12], which may have important consequences for the diagnosis, prognosis and treatment of the disease.

In this work we harness recent advances in DNA sequencing technologies using short and long-reads to characterise patterns of structural variation associated with HBV DNA integration in human HCC. Our analyses further illuminate a remarkable mutational mechanism, present in at least 8% of all HCC tumours, by which somatic integration of HBV DNA promotes non-homologous interchromosomal rearrangements coupled with megabase-size telomeric (i.e., that includes the telomere) deletions in one or two of the chromosomes involved, occasionally representing tumour driver events in HCC. We identify instances in which this process generates dicentric chromosomes, and removes relevant tumour suppressor genes in HCC evolution, such as *TP53*, *ARID1A*, *RB1*, *RPS6KA3* and *IRF2*. These events are clonal, and timing estimation reveals this mechanism is active in early stages of HCC evolution. Overall, these data underscore the importance of characterising liver cancer genomes for patterns of HBV integration, and provide insights for the prevention of the disease in a subset of HCC patients.

## Results

**Analysis of HBV integration sites identifies non-canonical HBV insertions.** We run our bioinformatic algorithms (Methods) to explore the landscape of HBV DNA integrations acquired somatically on Illumina paired-end whole-genome sequencing data from 296 HCC tumours from the Pan-Cancer Analysis of Whole Genomes (PCAWG) project[13]. Their matched-normal samples derived from blood, were also sequenced. This analysis retrieved a total of 148 somatic HBV integration events in 51 tumour samples (Fig. 1a and Supplementary Data 1). Forty-two of these events represent canonical viral DNA insertions where the paired-end mapping data shows a classical pattern, characterised by two reciprocal – face-to-face oriented – read clusters delimiting the integration site, and whose mates support the presence of viral DNA (Fig. 1b). This result is consistent with an alternative study on the same dataset carried out by others[14]. However, in addition to these canonical insertions, our analysis revealed that a majority (72%, 106/148) of events followed an unexpected, non-canonical pattern. Here, paired-end mapping data showed single clusters of reads whose mates identify one extreme of the somatic viral integration only, while the reciprocal cluster supporting the other extreme of the insertion appeared to be missing. For instance, in one HCC tumour, SA501453, paired-end reads show a single cluster supporting one extreme of an HBV insertion event on chromosome 19, with no reciprocal cluster in the proximity of the integration site (Fig. 1c). Our algorithms successfully reconstructed the ends of these 106 non-canonical insertion events, confirming that they match HBV sequences (Supplementary Data 2).

Similar paired-end mapping patterns were previously identified in cancer genomes with high retrotransposition rates[15], where this type of events represented hidden genomic rearrangements

mediated by aberrant DNA integrations. This suggested that our findings could represent cryptic somatic rearrangements mediated by HBV DNA insertion. Actually, somatic rearrangements linked to HBV insertion sites have been recently identified using long-read sequencing technologies in human HCC cell-lines[5] and primary tumours[10]. Hence, to illuminate the genuine configuration of the relevant rearrangement involved in the patterns described above, we performed long-read whole-genome sequencing on the affected tumour, SA501453, using Oxford Nanopore Technologies (ONT) to a final coverage of 13.5X (median read size = 12 kb). The long reads revealed a cryptic translocation linking chromosomes 19 and 11, which is bridged by a 640 bp HBV DNA insert (Fig. 2). Although our algorithms had initially identified the missing reciprocal cluster on chromosome 11 in the paired-end data, the interchromosomal rearrangement remained undetectable due to size constraints of the Illumina sequencing library, which was too short to span the HBV insertion. Notably, the genomic breakpoints of this translocation remained unnoticed to a set of four different structural variation calling pipelines, which were employed in the identification of genomic rearrangements and in the PCAWG dataset[13,16].

**Telomeric deletions mediated by HBV DNA insertions in HCC.** Many non-canonical HBV insertions occur in association with megabase-size deletions that remove telomeric regions of a chromosome. For instance, in HCC tumour SA529726, the paired-end sequencing data revealed one single cluster of an HBV insertion on the short arm of chromosome 3. Here, the insertion boundary is associated with a large copy number loss (Fig. 3a), suggesting that the insertion event occurred in conjunction with a telomeric deletion that removed 21 Mb of chromosome 3p. We performed long-read sequencing on this sample, which revealed that the telomeric deletion occurred due to an unbalanced translocation between chromosomes 3 and X bridged by a 3.3 kb HBV insertion that shows a classical fragmented and rearranged form[5,17] (Fig. 3a and Supplementary Fig. 1a). In the same sample, the ONT data showed a second, unrelated HBV insertion (3.5 kb long) that bridges a translocation between chromosomes 4 and 7, associated with a 20 Mb telomeric deletion on 4q (Fig. 3a and Supplementary Fig. 1b). Similarly, in another remarkable HCC tumour, SA501511, up to three different HBV insertions were found associated with large deletions – 20.5, 33.6 and 76.7 Mb long – removing the telomeres on chromosome arms 10p, 4p and 13q, respectively (first circos plot in Fig. 3b). This time, the long-read sequencing data revealed three cryptic HBV-mediated translocations between the long arm of chromosome 8 and the relevant deletion breakpoints on chromosomes 4, 10 and 13 (Fig. 3b and Supplementary Fig. 2a–c).

We looked in the PCAWG HCC dataset for other HBV insertions demarcating huge telomeric copy number loss events, which could involve the same mutational mechanism, finding 26 additional events in 19 different HCC tumours (Supplementary Fig. 3). Hence, we find that ~8% (23/296) of all HCC samples in the PCAWG dataset bear the hallmarks of this mutational process (Supplementary Data 1). These 23 samples bear in total 40 telomeric deletions apparently caused by HBV integration. We analysed three of these samples (SA501424, SA501481 and SA529830) by whole-genome long-read sequencing with ONT, which confirmed cryptic interchromosomal rearrangements linked to telomeric deletion breakpoints in all of them (Fig. 3b), demonstrating that this aberrant mutational mechanism mediated by HBV insertions is recurrent in human HCC. Notably, in two of the samples sequenced with ONT (SA501481 and SA529830) the configuration of the rearrangements found supports a derived chromosomal fusion that generates a

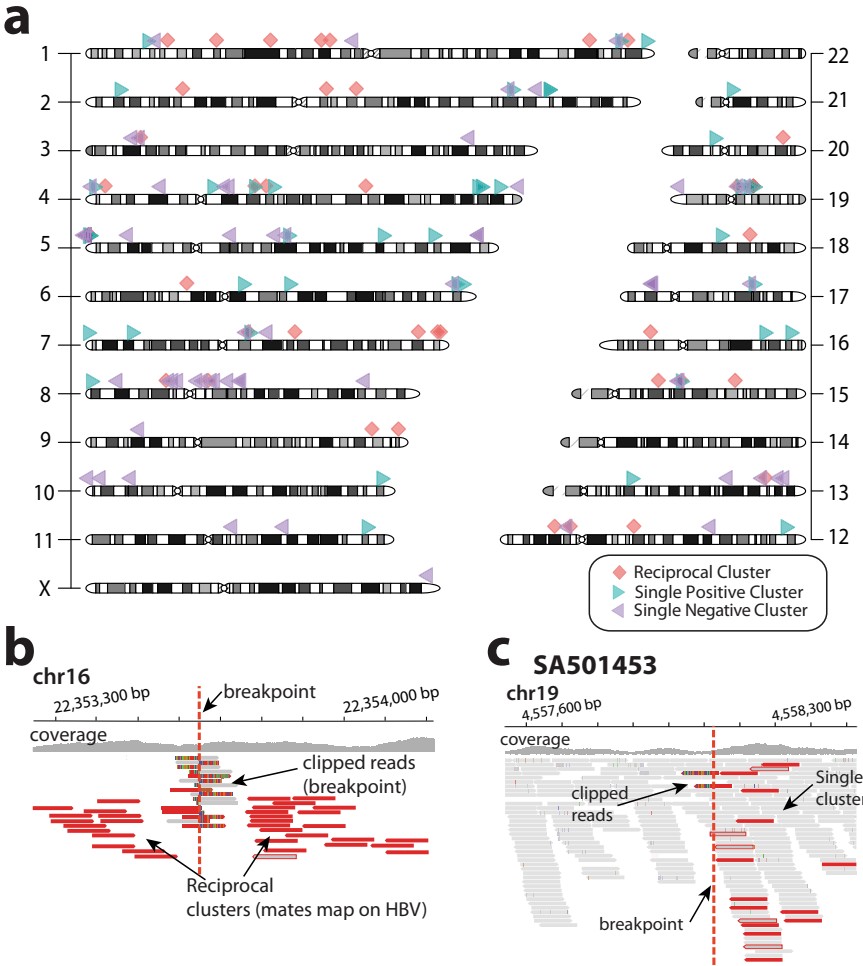

**Fig. 1 The landscape of HBV insertions in 296 HCCs from the PCAWG dataset. a** Canonical (reciprocal) insertions are represented as red diamonds, and non-canonical insertions (single-clusters) as purple and green triangles for positive and negative clusters, respectively. In total, 148 integration events are shown of which 72% represent non-canonical events. **b** Classical pattern of canonical HBV insertions identified with Illumina paired-end mapping data is characterised by two reciprocal clusters of discordant reads, and clipped reads, in face-to-face orientation, demarcating the boundaries of the genomic integration. The mates of these reads map onto HBV consensus sequences. Clipped reads span the insertion site allowing base-pair resolution of the insertion breakpoints. **c** Most HBV insertion events in HCC tumours show a non-canonical pattern in which a single cluster of paired-end reads (short-reads in red) demarcates one of the two boundaries of the insertion only, while the second cluster is missing.

dicentric chromosome (i.e., a chromosome with two centromeres; Fig. 3b). These chromosomes are known to represent a potential source for breakage-fusion-bridge (BFB) repair[15,18], unless they become stabilised due to reduced intercentromeric distance or by means of inactivation of one of the two centromeres[19]. Here, the absence of copy number profiles and chromosomal rearrangements typically associated with BFB cycles supports the last scenario. In this work we obtained the whole genomes from a total of 9 human of HCC samples in the PCAWG dataset using ONT long-read sequencing, 7 of them bearing HBV-mediated telomeric deletions. The reconstruction of the nucleotide sequences from the inserted HBV DNA shows that, in general, insertions mediating telomeric deletions tend to have a more complex structure than canonical ones, including duplications and inversions of the reference HBV genome, including internal deletions, inversions and duplications (Supplementary Fig. 4).

**HBV insertion rate varies across the HCC genome.** To understand the properties of the insertion points of HBV DNA, we analysed the genome-wide distribution of 148 somatic events (including 42 canonical and 106 non-canonical). We find considerable variation in the rate of HBV integration (Fig. 4a). One region at chromosome 5p, with 13 HBV insertions, stands out over the others, with a hot-spot in the *TERT* gene (11 HBV insertions in *TERT*, 9 of them located upstream to the gene close to the promoter region; Fig. 4a and Supplementary Data 3). This cancer gene has been previously identified as a main target for HBV insertions[10,11,20,21], which are the cause of *TERT* focal amplification and *TERT* promoter activation[10]. Notably, we find that 72% (8/11) of the events in *TERT* represent non-canonical insertions of HBV, which confirms previous data showing that this relevant gene can be targeted by genomic rearrangements bridged by the virus[10]. Other cancer-related genes with recurrent insertions within gene boundaries are *KMT2B* (n = 4, three in coding exons) and *CCNE1* (n = 2, all intronic), both at chromosome 19q (Fig. 4a).

To further characterise the patterns of HBV integration in human HCC, we studied the association of HBV events and genomic features, including chromatin state, replication timing and gene expression (Fig. 4b, c and Supplementary Fig. 5). This analysis shows a significant enrichment of HBV events in genes (P = 0.035, Fig. 4b) and a depletion in intergenic regions (P = 0.035; Fig. 4b). Notably, 61% (91/148) of all HBV events fall within gene boundaries. When we split the HBV events into those linked to megabase-size telomeric deletions (n = 40) and

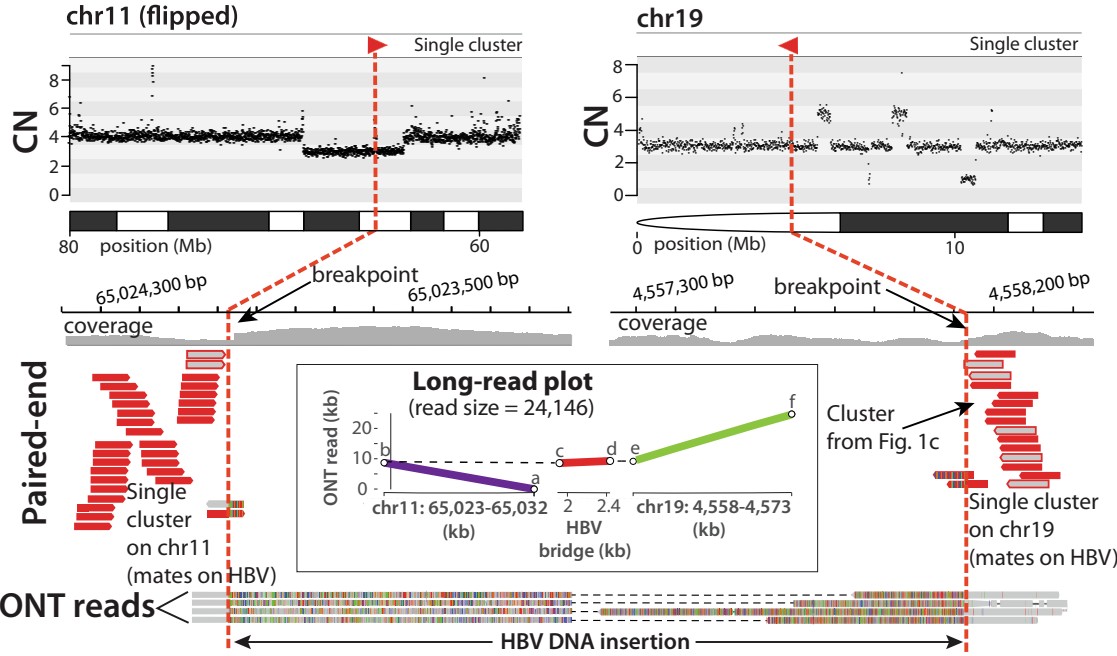

**Fig. 2 Long-read sequencing reveals cryptic HBV-mediated translocations in human HCC.** In HCC SA501453, a hidden interchromosomal rearrangement between chromosomes 11 and 19 is identified using Oxford Nanopore Technologies (ONT). The copy number plot (CN) at the top shows the copy number profiles of the chromosomes involved in the rearrangement (note that the CN plot on chromosome 11 is flipped for illustrative purposes). The Illumina paired-end sequencing data (short-reads in red) shows two single clusters of discordant read pairs, one on 11q and a second on 19p, pointing to HBV insertion events that cannot be bridged due to Illumina library size constrains. The bottom shows four long-reads obtained with ONT that reveals the real configuration of the hidden rearrangement, consisting of a 640 bp HBV DNA insertion bridging a translocation between 11q and 19p. ONT reads were cut (discontinued) for illustrative purposes. The long-read plot represents the alignment of one ONT read – 24 kb long – to chromosomes 11 and 19 of the human reference genome and to an HBV consensus sequence. The long-reads used to construct the long-read plots are annotated in Supplementary Table 1.

the remaining HBV insertions ($n = 108$), we observe that while the category of HBV insertions with no telomeric deletions exhibits the expected pattern for canonical HBV insertions, characterised by an enrichment in regions of early replication timing[20], HBV insertions causing telomeric deletions are depleted in those regions ($P = 0.013$ and $P = 0.037$, respectively; Fig. 4c). This could obey to different DNA repair mechanisms driving the integration of HBV DNA at different stages of S phase[22].

**HBV insertions are clonal events acquired early in HCC evolution.** Our results illuminate a scenario where rearrangements mediated by viral DNA integration are important remodelers of human HCC genomes. The analysis of copy number profiles revealed that many HBV-mediated rearrangements occurred in chromosomes with copy number gains, providing opportunities for timing analyses[23,24]. To pinpoint these rearrangements on a timeline from the fertilised egg to tumour diagnosis, we modified current timing algorithms to operate with single read-clusters only (see Methods section). The method revealed that somatic insertions of HBV DNA in HCC are typically clonal events that have been acquired early in tumour evolution (i.e., prior to the copy number gain). For example, in one notable HCC, SA269680, which underwent whole-genome duplication (WGD), we identified eleven viral insertion events. All but one were catalogued as early events (Fig. 5a and Supplementary Data 4), and five of these early events corresponded to single clusters associated with megabase-size copy number losses (Supplementary Fig. 3), supporting the notion that these large-scale rearrangements may be important in the initial stages of liver oncogenesis.

To further investigate the clinical relevance of HBV integration in HCC, we employed real time estimation data of WGD events from PCAWG[23] to perform a more precise timing estimation of HBV events along patients' lifetime. The method is based on the analysis of mutational clock signatures that correlate with patient age at diagnosis[25], which can be used for timing of WGDs and their associated variants[23]. This approach allowed real-time timing of 37 HBV insertions (8 canonical and 29 non-canonical) embedded in WGD tumours (Fig. 5b and Supplementary Data 5), and revealed some of these rearrangements appear many years before diagnosis. For instance, in HCC SA501645, a cryptic HBV-mediated rearrangement in chromosome 10, coupled with a 7.3 Mb telomeric deletion on 10p, occurred over 21 years before the patient was diagnosed with HCC (Fig. 5b, c).

**HBV-mediated telomeric deletions cause loss of tumour suppressor genes.** We find 40 HBV-mediated rearrangements with telomeric deletions involving 23 HCC samples. Notably, there are instances in which essential tumour suppressor genes are lost by this mutational mechanism, which may provide the driver rearrangements that a cancer clone requires to survive and grow. In one remarkable HCC, SA529830, we identified one paired-end single cluster supporting an HBV insertion on the short arm of chromosome 17. The insertion occurred in conjunction with a 14.9 Mb clonal telomeric deletion at the integration site, which removed one copy of tumour suppressor gene *TP53* (Fig. 6). Notably, the second copy of *TP53* in this tumour is inactivated by the missense point mutation C242S[26,27] (Supplementary Fig. 6). The paired-end data showed a similar pattern on the short arm of

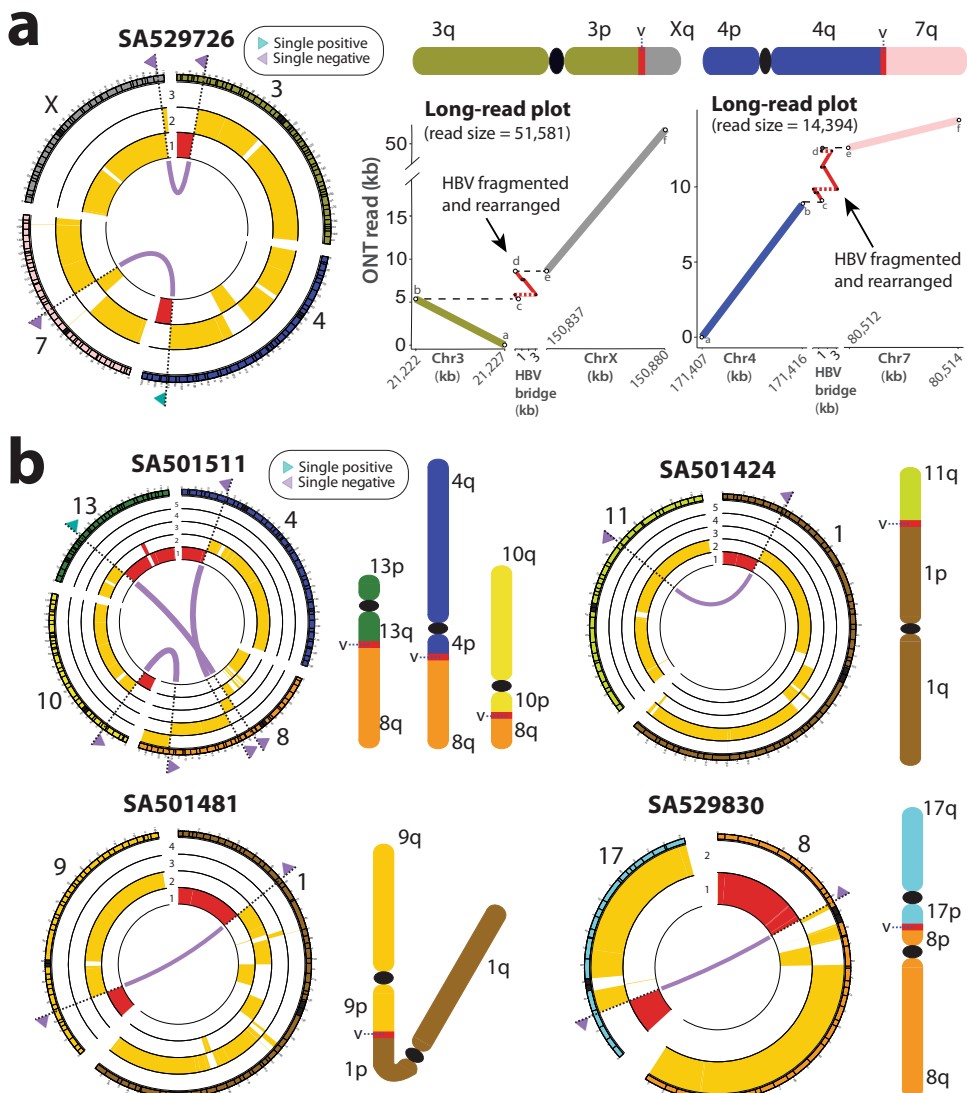

**Fig. 3 HBV DNA integration mediates interchromosomal genomic rearrangements that lead to megabase-size telomeric deletions in HCC. a** In tumour SA529726, two unrelated HBV-mediated interchromosomal rearrangements between chromosomes 3 and X, and between chromosomes 4 and 7, promote 21.2 Mb and 19.8 Mb telomeric deletions on the 3p and the 4q, respectively. The circos plot (left) represents the translocations (purple lines) revealed by ONT data. Single clusters identified with paired-end mapping data are denoted as triangles (green for positive orientation, purple for negative) on the chromosome ideograms. The copy number profiles are shown in yellow below the chromosome ideograms, with relevant telomeric deletions highlighted in red. The long-read plots (right) represent the alignment of one ONT long-read to chromosomes 3 and X, and chromosomes 4 and 7, of the human reference genome and an HBV consensus sequence, which validates the interchromosomal rearrangements mediated by the virus shown in the circos plot. Here, the analysis of the long-reads supporting the HBV events showed an HBV DNA insertion in a classical fragmented and rearranged form[5, 17]. The expected configuration of the rearranged chromosome is shown above each long read plot (the ideograms are for illustrative purpose only); 'v' denotes the HBV insertion. The long-reads used to construct the long-read plots are annotated in Supplementary Table 1. **b** Circos plots and chromosome diagrams of similar HBV-mediated non-homologous translocations promoting megabase-size telomeric deletions in four additional HCC tumours. Again, the expected configuration of the rearranged chromosome is shown next to each circos (the ideograms are for illustrative purpose only). In SA501511, three unrelated HBV-mediated translocations involving different loci on chromosome 8 promote huge deletions involving telomeric regions on chromosomes 13q, 4p and 10p. In SA501424, one HBV insertion bridges a genomic translocation between chromosomes 1 and 11 that generates a terminal deletion at 1p. In SA501481 and SA529830, HBV-mediated translocations generate dicentric chromosomes and promote megabase-size terminal deletions.

chromosome 8, where a single cluster supporting an HBV insertion occurred together with a loss of the first 41 Mb of the chromosome. The patterns suggested that an HBV DNA molecule could be bridging an unbalanced translocation between chromosomes 17 and 8 that would generate a dicentric chromosome (see circos plot in Fig. 3b). We carried out whole-genome long-read sequencing, which confirmed the expected configuration of this relevant rearrangement (long-read plot in Fig. 6). In addition, we performed in-situ hybridisation to identify

the loss of *TP53* and the chromosomal fusion between chromosomes 17 and 8, which further validated these concomitant events (Supplementary Fig. 7).

Similarly, in one additional HCC tumour, SA501481, we identified an HBV insertion into chromosome 1 associated with the deletion of one copy of tumour suppressor gene *ARID1A* (Fig. 7a). Here, paired-end data shows a single cluster of reads, whose mates support the HBV insertion, demarcating a copy number loss of the first 57.2 Mb of 1p including *ARID1A*. Again,

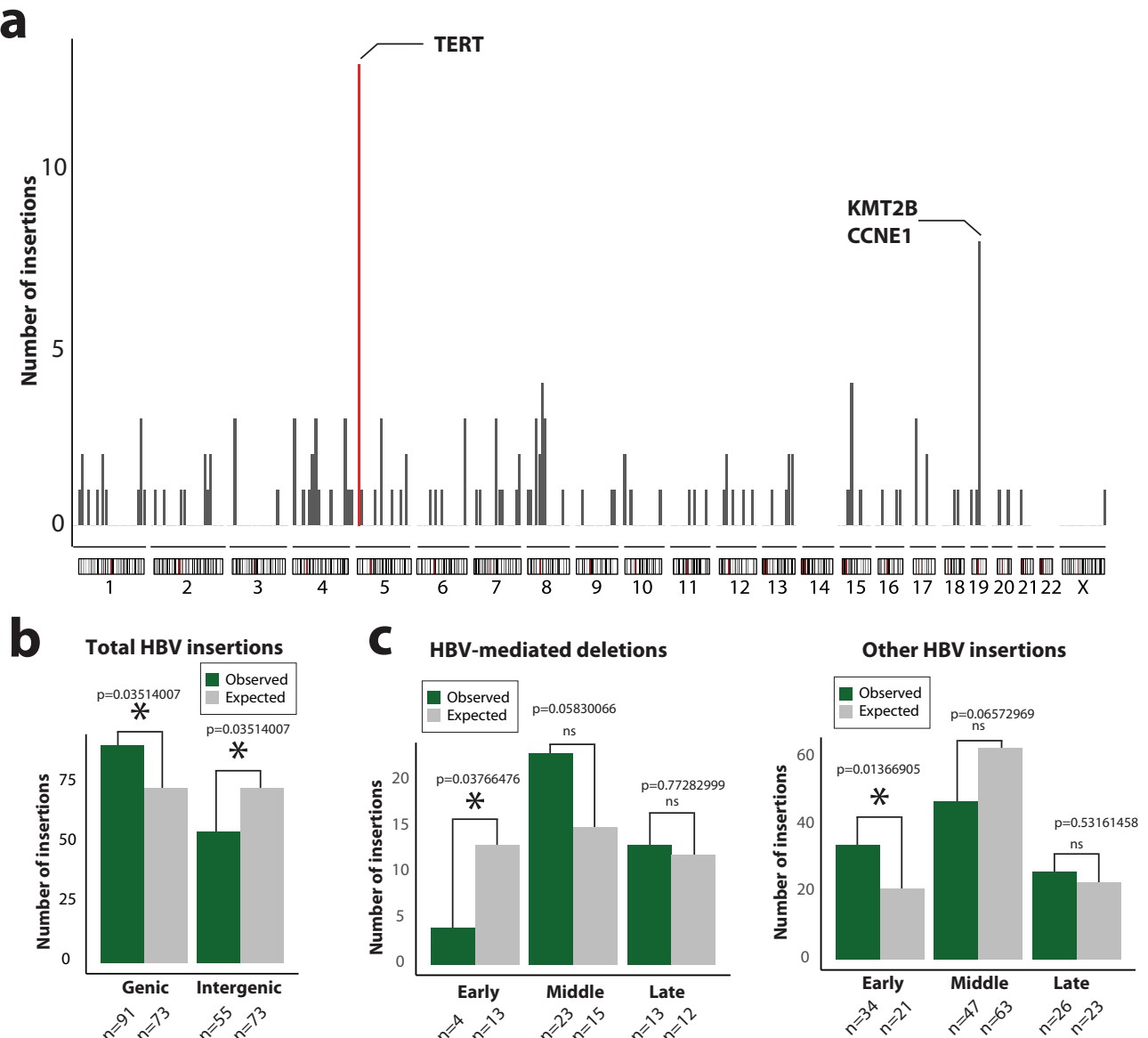

**Fig. 4 Distribution of HBV insertions across the HCC genome and association with genomic organisation features. a** Bars show the number of HBV insertions per 10-Mb window of the genome. Red bar represents the *TERT* hotspot at chromosome 5p. Other genes (*KMT2B* and *CCNE1*) with high rate of HBV insertion are shown at chromosome 19q. **b** HBV insertions are more frequent in genes than expected by chance ($\chi^2$ test, $P = 0.0035$, $n = 91$) and depleted in intergenic regions of the genome ($\chi^2$ test, $P = 0.035$, $n = 55$). False Discovery Rate "FDR" correction was applied (see Methods). **c** HBV events at telomeric deletion breakpoints are depleted in regions of the genome exhibiting early replication timing ($\chi^2$ test, $P = 0.037$, $n = 4$), while the remaining HBV insertions show the opposite pattern ($\chi^2$ test, $P = 0.013$, $n = 34$). "ns" stands for not significant. FDR correction was applied.

in this case, we initially lacked the DNA region on the other side of the rearrangement mediated by the virus, due to Illumina library insert size constraints. The paired-end data showed an analogous pattern in chromosome 9, with an independent cluster supporting an HBV insertion that occurred together with a telomeric deletion of the first 41 Mb of 9p at the integration site. This scenario suggested a cryptic unbalanced translocation between 1p and 9p, generating a dicentric chromosome (see circos in Fig. 3b), which was confirmed by long-read sequencing with ONT (long-read plot in Fig. 7a).

*ARID1A* is a relevant cancer gene harbouring monoallelic loss-of-function mutations in 10-15% of human HCC samples[28]. Notably, in a different HCC, SA501424, we found a similar scenario to the one described above. This time, an HBV insertion demarcates a deletion of the first 31.5 Mb of chromosome 1p,

which again involved loss of one copy of *ARID1A* (Fig. 7b). Hence, we performed long-read sequencing with ONT, which revealed a cryptic interchromosomal rearrangement between chromosomes 1p and 11q bridged by HBV (see circos in Fig. 3b). The deletion and the chromosomal fusion were also validated by in-situ hybridisation (Supplementary Fig. 7).

Overall, the analysis of the copy number data from the 23 HCC samples with telomeric deletions show that these alterations result in the loss of 244 genes classified as tumour suppressor or loss-of-function genes, according to Cancer Gene Census[29] and Intogen[30], respectively (Supplementary Data 3). Thirty-seven of these genes are found to be involved in human HCC in the mentioned databases. Notably, at least three of these events are reported in the Compendium of driver copy number alterations of the PCAWG Consortium[13]. These three events are described

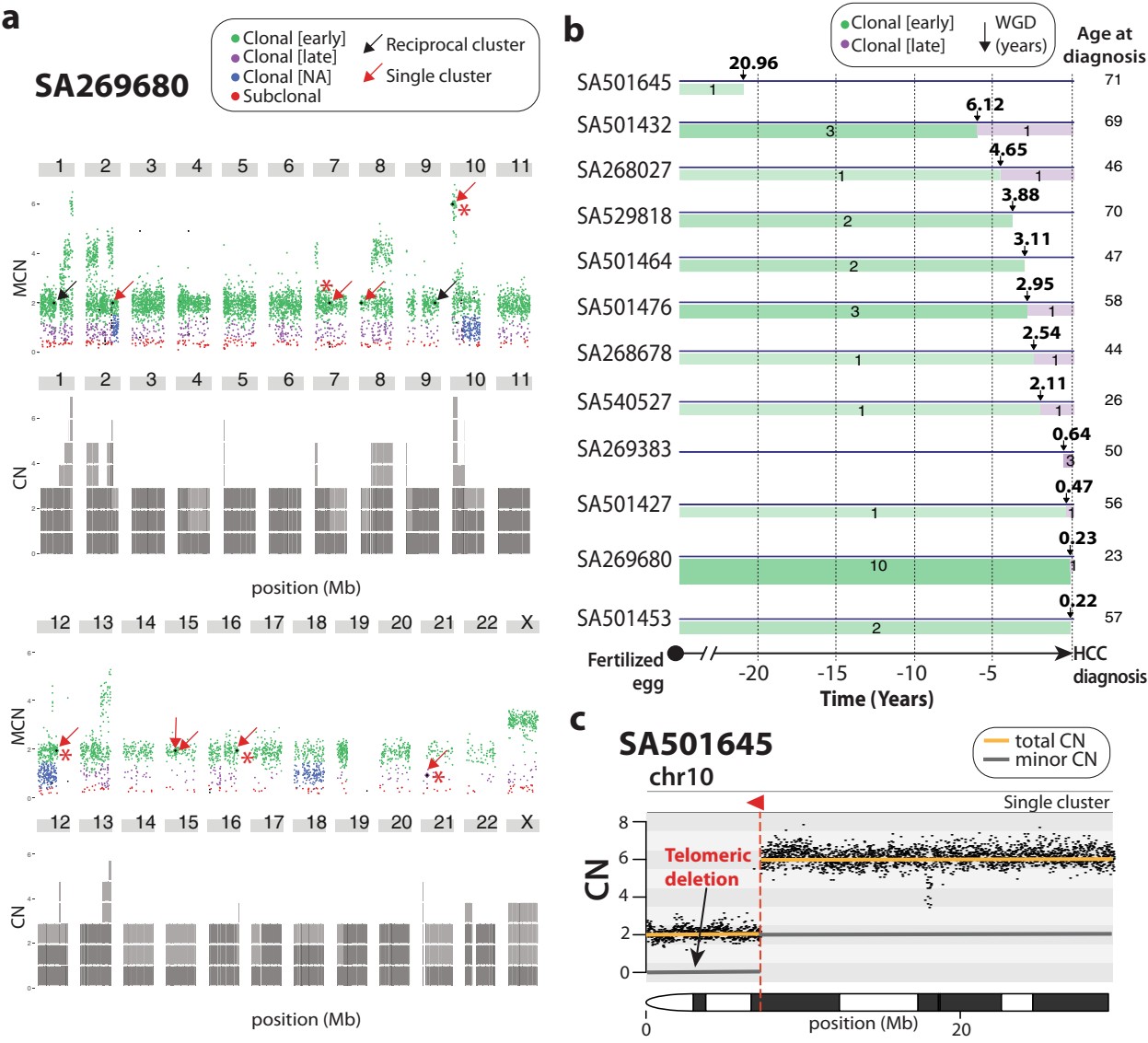

**Fig. 5 HBV-mediated rearrangements are early clonal events in HCC evolution. a** In SA269680, an HCC with a whole-genome duplication, HBV insertions are shown in the context of point mutation burden for that sample. Coloured dots above chromosomes represent point mutations with different timing: early clonal (before the whole-genome duplication; green), late clonal (after the whole-genome duplication; purple), clonal (blue), subclonal (red). We identified nine HBV single clusters (black dots with red arrows), all but one catalogued as early clonal events. Five of these early HBV insertions (marked with red asterisks) are associated with megabase-size telomeric deletions (see copy number plots in Supplementary Fig. 3). The same sample bears two additional early clonal HBV canonical insertions (black dots with black arrows). Grey blocks below chromosomes represent the copy number profile. MCN stands for Minor Copy Number. **b** Real-time timing estimation of HBV insertions along patients' lifetime in samples with whole-genome duplication events. The X axis shows the time interval when – before (green) and after (purple) – the somatic HBV insertions took place relative to the WGD event; thickness and strength of the green and purple bars correlates with the number of events. Black arrows represent when a WGD event took place, and numbers above arrows show the time – in years – before HCC diagnosis when the WGD event has occurred. Numbers within green and purple timelines represent the number of insertion events. Numbers at the end of the timeline represent the age of the patient at diagnosis. **c** Copy number plot showing a single cluster that supports an HBV insertion event (red triangle) associated with a 7 Mb telomeric deletion on chromosome 10 in SA501645 that, according to Fig. 4b, occurred at least 20.96 years before HCC diagnosis. The gold line represents total chromosome CN, and the grey line is the minor chromosome CN.

above and include the deletion of *TP53* in sample SA529830 (Fig. 6), the deletion of *ARID1A* in sample SA501424 (Fig. 7), and the loss of one copy of the *RB1* gene in a 13q telomeric deletion from sample SA501511 (Fig. 3b and Supplementary Fig. 2c). In addition, although not included in the PCAWG Compendium, we find three additional clonal telomeric deletions removing one copy of two relevant cancer genes in human HCC. First, in sample SA529726, a telomeric deletion at chromosome X removes one

copy of the loss-of-function gene *RPS6KA3*[31] (this rearrangement is described in Fig. 3a). Second, we find two unrelated deletions, in patients SA268027 and SA269383, that remove one copy of the *IRF2* gene (Supplementary Fig. 3). Although *IRF2* is not catalogued in Census nor Intogen, functional studies have identified it as a cancer suppressor gene in HCC[31]. These results provide evidence that this is a mutational mechanism that likely contributes to the development of human HCC.

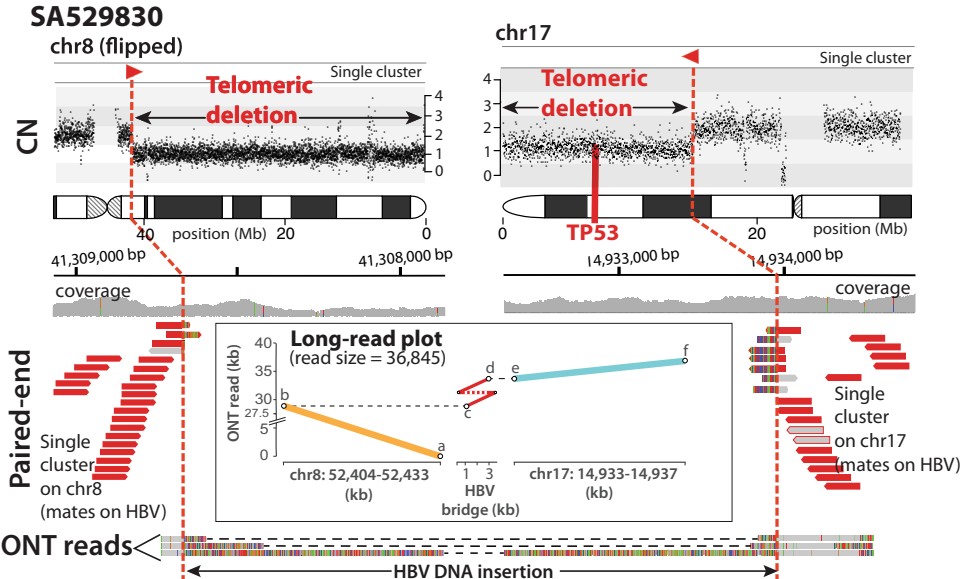

**Fig. 6 HBV-mediated translocations may lead to loss of tumour suppressor gene *TP53*.** In HCC SA529830, a cryptic interchromosomal rearrangement between chromosomes 17 and 8 is bridged by a 4829 bp HBV insertion associated with a 14.9 Mb telomeric deletion on chromosome 17 that removes one copy of the tumour suppressor gene *TP53*, and a second 43 Mb telomeric deletion on chromosome 8. Note that the CN plot on chromosome 8 is flipped for illustrative purposes. Single paired-end clusters (short-reads in red) on chromosomes 17 and 8 demarcate the boundaries of both deletions and support the insertion of HBV DNA. One ONT read of 36,845 bp evidences the extent of the rearrangement, whose alignment onto the reference genome – chromosomes 8 and 17 – and to a consensus HBV sequence is shown in the long-read plot. ONT reads were cut (discontinued) for illustrative purposes. The configuration of the rearrangement predicts the formation of a dicentric chromosome (Fig. 3b). The long-reads used to construct the long-read plots are annotated in Supplementary Table 1.

## Discussion

Most cancers are characterised by somatic acquisition of genomic rearrangements during tumour evolution that, eventually, drive the oncogenic process[32]. These structural aberrations are caused by different mutational mechanisms that generate particular patterns or signatures in the DNA[33]. Identification of these mechanisms and their associated patterns is necessary to understand the dynamic processes shaping the cancer genome. Here we described the patterns of a recurrent, quite remarkable mutational mechanism occurring in the early stages of human HCC development whereby HBV DNA integration mediates interchromosomal rearrangements contributing to megabase-size telomeric deletions, which may lead to loss of tumour suppressor genes. Our results demonstrate that the consequences of this mutational mechanism are dramatic for the architecture of HCC genomes and, on occasion, the resulting structural configuration can drive the oncogenic process. We have used two (related) ways of estimating the timing of HBV insertions and their mediated genomic rearrangements, which demonstrate this mutational mechanism is active early during tumour evolution and show that some HBV-mediated rearrangements can occur as early as 21 years before cancer diagnosis. Overall, these data underscore the importance of characterising liver cancer genomes for patterns of HBV integration, and raise the question about the potential benefit of an earlier antiviral therapy against HBV, in HBV-infected patients, to prevent the acquisition of early driver mutations caused by the virus in the initial stages of liver cancer development. This is particularly relevant for chronic hepatitis B patients in the "immune-tolerant" phase, a stage of the disease characterised by no-fibrosis or minimal fibrosis for which antiviral therapy is not recommended by current patient management guidelines[34–36]. However, levels of HBV replication in this stage are very high[34,35], generating the template double-strand linear DNA (dslDNA)[37] required for the somatic acquisition of the HBV DNA insertions and HBV-mediated rearrangements that, eventually, drive the oncogenic process.

## Methods
### Materials and methods

*Sequencing dataset and DNA sources.* We analysed Illumina whole-genome paired-end sequencing reads, 100–150 bp long, from 286 hepatocellular carcinoma (HCC) tumours and their matched-normal samples from the Pan-Cancer Analysis of Whole Genomes (PCAWG) project dataset[38]. The tumour specimens consisted of a fresh frozen sample, while the normal specimens consisted of a blood sample. The average coverage was 44 reads per bp for tumour samples and 33 reads per bp for matched normal samples. BWA-mem algorithm v0.78.8-r455[39] was used to align sequencing reads to human reference genome build GRD37, version hs37d5. Tumour's DNA for additional long-read sequencing and FISH were transferred from the HCC tumours collection at RIKEN (Japan) within the framework of the International Cancer Genome Consortium (IGCG). The ethics oversight for the PCAWG protocol was undertaken by the TCGA Program Office and the Ethics and Governance Committee of the ICGC. Each individual ICGC and TCGA project that contributed to the PCAWG dataset had their own local arrangements for ethics oversight and regulatory alignment. We performed additional whole-genome sequencing with Oxford Nanopore (ONT) on nine HCC tumours from the PCAWG project (all are ICGC tumours only). The tumour specimens consisted of a fresh frozen sample. The average ONT sequencing coverage was 8 reads per bp (range 3.83–13.97), and the mean read length was 7.9 kb. All tumours sequenced and analysed in this project have written informed consent for the usage of DNA samples for whole-genome sequencing (for either short-reads or long-reads) and for publication of their genomic data. All patients agreed to participate in the ICGC study and provided informed consent following ICGC guidelines.

*Detection of viral insertions using v-TraFiC.* v-TraFiC represents a modified version of former algorithm TraFiC[40], for the identification of somatic insertion events of viral DNA using paired-end sequencing data in three main steps: (i) selection of candidate reads; (ii) reads clustering; and (iii) identification of viral DNA events.

(1) Candidate reads selection
v-TraFiC (v0.23) identifies reads from BWA-mem mapping that are likely to provide information pertaining to viral DNA site inclusion. Two different read-pair types are considered for the identification of viral insertions, named SINGLE_END (i.e., one end of the pair – called anchor is mapped onto the reference genome while the other is unmapped), and ABERRANT (i.e., both reads of the pair are improperly mapped to a chromosome, where the read with the highest MAPQ is considered the anchor). In both cases, the anchor's MAPQ must be higher than zero, its mapping pattern must not be 'soft clipping–alignment match–soft clipping' (i.e., CIGAR string must not be #S#M#S, where # represents the number of nucleotides), and must not map onto decoy sequences, mitochondrial DNA or Y chromosome. In

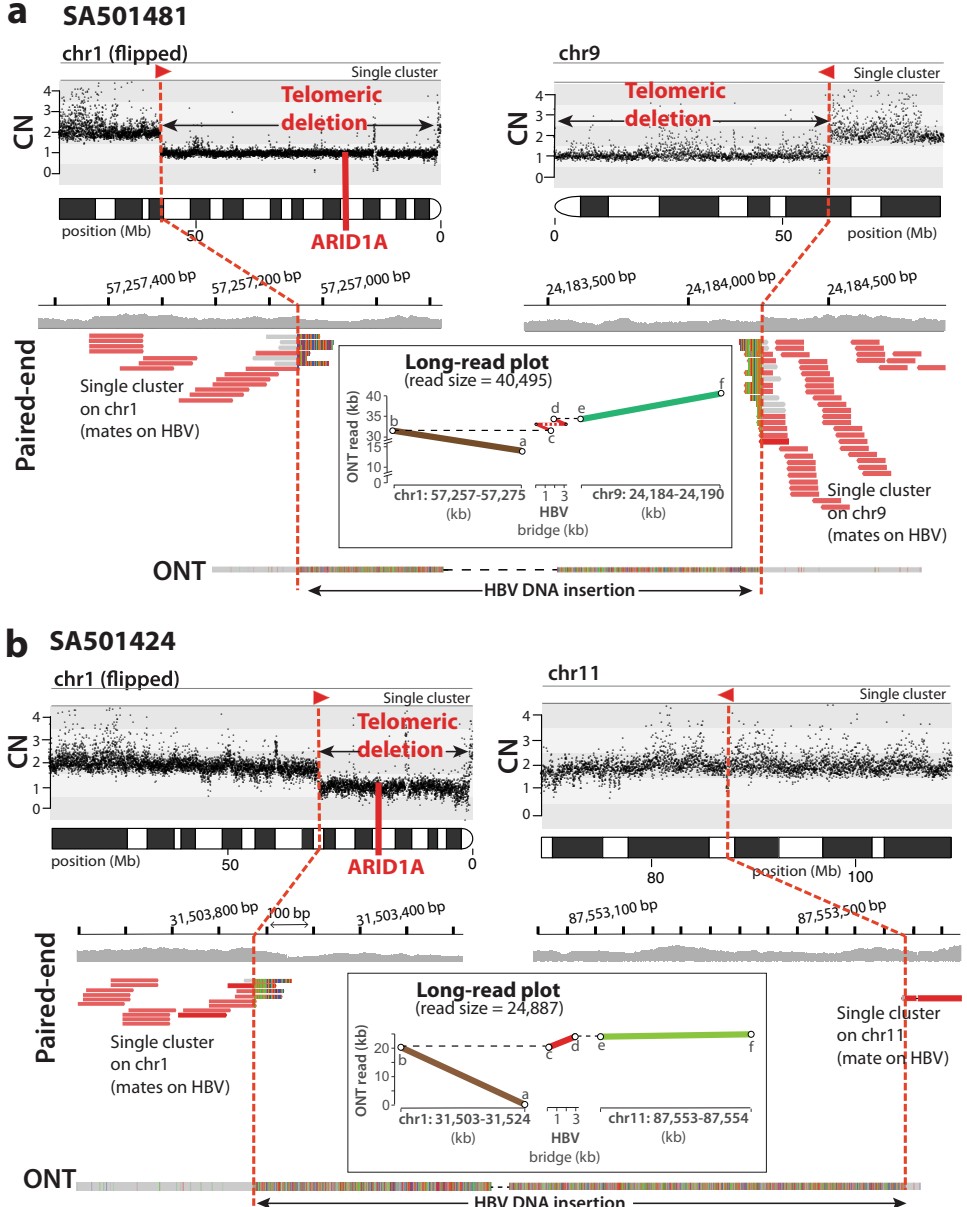

**Fig. 7 HBV-mediated translocations lead to recurrent loss of tumour suppressor gene *ARID1A*. a** In HCC tumour SA501481, the Illumina paired-end data (short-reads in red) shows two clusters, one on chromosome 1 and another on chromosome 9, which point to both extremes of an HBV insertion. The copy number (CN) plot at the top shows the total (gold line) and minor (grey line) chromosomes' copy number profiles. The CN plot reveals two telomeric deletions associated with HBV events, one that removes 57.2 Mb on 1p, including one copy of the *ARID1A* tumour suppressor gene, and a second deletion that removes 21.2 Mb on 9p. Note the CN plot from chromosome 1 is flipped for illustrative purposes. The long-read plot shows a 2688 bp HBV insertion that bridges an interchromosomal rearrangement between chromosomes 1p and 9p. The configuration of the rearrangement predicts the formation of a dicentric chromosome (Fig. 3b). **b** A similar scenario, in tumour SA501424, where an HBV DNA insertion induces an interchromosomal translocation between chromosomes 1 and 11. The Illumina paired-end data (short-reads in red) shows two single clusters, one on chromosome 1 and another on chromosome 11, which point to both extremes of an HBV insertion. The CN plot at the top reveals a 31.5 Mb telomeric deletion on 1p associated with the HBV insertion event (note that the CN plot from chromosome 1 is flipped for illustrative purposes). Here, the associated telomeric deletion on chromosome 1 removes one copy of tumour suppressor gene *ARID1A*. The long-read alignment plot demonstrates an interchromosomal rearrangement between chromosomes 1 and 11 mediated by an HBV insertion. The long-reads used to construct the long-read plots are annotated in Supplementary Table 1.

addition, the pair is also excluded if any of the reads is not a primary alignment, fails platform/vendor quality checks, or is PCR or optical duplicate. Non-anchor reads must not contain unsequenced nucleotides ('N') and MAPQ of non-anchor ABERRANT reads must be < 20. The algorithm dustmasker v2.6.0[41] is used to identify non-anchor read-pairs containing low complexity sequences, which are later discarded.

(2) Clustering

Anchor reads are clustered together if (i) they share the same orientation, and (ii) the distance relative to the nearest mapped read of the same cluster

is ≤200 bp. Two main cluster categories are defined, namely POSITIVE and NEGATIVE (i.e., anchor reads are mapped onto the positive and negative strand, respectively). A preliminary range of genome coordinates is associated with each single cluster – final breakpoint coordinates are refined in a later step –. Ranges are defined by a lower (left) coordinate (P_L_POS and N_L_POS, respectively for positive and negative clusters) and an upper (right) coordinate (P_R_POS and N_R_POS, for positive and negative). Only clusters consisting of ≥4 supporting reads are considered for further analysis. To avoid miscalls due to alignments in complex regions, the full set

of reads mapping within cluster coordinates [P_L_POS, N_R_POS] are further analysed, and clusters are removed if: (i) the proportion of reads with MAPQ ≤ 10 relative to the total reads mapped within cluster boundaries represents >0.3 (30%), and/or (ii) the proportion of reads with CIGAR string #S#M#S relative to the total reads mapped within cluster boundaries represents <0.15 (15%). Clusters in the tumour are removed if a syntenic cluster in the matched-normal sample is detected with the same orientation and mapping the same locus <500 bp away. Finally, one positive and one negative clusters are reciprocal if P_R_POS ≥ N_L_POS and abs(N_L_POS − N_R_POS) ≤ 350 bp, otherwise clusters are catalogued as single (or independent).

(3) Identification of viral DNA events

Non-anchored reads from each cluster were de novo assembled using Velvet v1.2.10[42], and contigs were used as queries of BLAST v2.6.0 searches against the RVDB Reference viral database[43] v12.2 containing 2,467,269 viral DNA sequences, of which 91,455 correspond to human Hepatitis B virus (HBV). Only contigs matching human HBV DNA are considered, and reciprocal clusters pointing to HBV DNA are catalogued as canonical HBV DNA insertion events, while single, independent clusters are catalogued as candidates for aberrant HBV DNA integration events. Finally, we used the algorithm MEIBA v.0.8.8[15], to identify and reconstruct HBV DNA insertion breakpoints to base-pair resolution, with the following non-default parameters: 'Maximum number of clipped read clusters in the insertion region' = 20 (default = 10), and 'Window size to search for clipped read clusters from discordant read-pair clusters ends' = 100 bp (default = 50 bp).

*Identification of HBV-mediated translocations and validation of v-TraFiC calls using single-molecule sequencing with Oxford Nanopore.* We performed long-read whole-genome sequencing with Oxford Nanopore Technologies (ONT) on nine native HCC tumours with relevant HBV DNA insertion events (i.e., SA501491, SA529726, SA529759, SA529830, SA501424, SA501453, SA501481, SA501511, SA501534). Libraries were constructed using the Oxford Nanopore Sequencing 1D ligation library preparation kit (SQK-LSK109, Oxford Nanopore Technologies Ltd) according to the manufacturer's protocol, including an initial DNA repair step with NEBNext FFPE DNA Repair Mix (New England BioLabs) and NEBNext Ultra II Ligation Module (New England BioLabs). Two low DNA yield samples (SA529726 and SA501481) were whole-genome amplified using φ29 DNA polymerase (REPLI-g midi kit, Qiagen) prior library construction. Amplified DNA was then digested with t7 endonuclease I (New England BioLabs) for linearization of branched amplicons and deproteinized with Proteinase K (New England BioLabs). Next, unbranched DNA underwent size selection of fragments longer than 20 Kb by means of a Short Read Eliminator buffer (Circulomics) precipitation step and was further purified with Ampure XP Beads (Beckman Coulter Inc). Then, libraries were obtained according the manufacturer´s protocol as described above.

Sequencing was performed onto MinION R9.4 flowcells (FLO-MIN106 rev-D, Oxford Nanopore Technologies Ltd), controlled by the Oxford Nanopore MinKNOW software v18.12.09 to v19.12.5. Base-calling and post-processing of the ONT raw fast5 files was conducted with ONT software Albacore v2.3.4 or Guppy v2.3.1 to obtain fastq files. Files with quality scores below the recommended values were dropped at this point from further analysis. Reads for each library were then independently mapped to the hs37d5 human reference genome with minimap2 v2.14-r883[44] and the resulting SAM files were converted to BAM files, sorted and indexed using Samtools v1.7[44]. All partial BAM files were merged, sorted and indexed to the final BAM files.

We performed validation of 47 putative somatic HBV insertion events (36 single clusters and 11 reciprocal insertions) identified with v-TraFiC in the 9 HCC tumours that were sequenced using Illumina paired-end and ONT long-reads. For each one of the HBV events we interrogated the long-read tumour BAM file to seek for long-reads validating the event. Two types of supporting reads were employed, namely (i) 'spanning-reads', composed of ONT reads completely spanning the HBV insertion, hence they can be identified as a standard insertion on the reference genome, and (ii) 'clipped-reads', composed of ONT reads spanning only one of the HBV insertion ends, hence they get clipped the alignment onto the reference genome. HBV events supported by at least one ONT read were considered true positive events, while those not supported by such reads were considered false positive calls. Overall, we find ~10% (5/47) of false positive events (note that this rate could be overestimated due to low coverage in the ONT data). Spanning-reads were used to identify 11 cryptic translocations.

*Copy-number dataset.* We analysed copy number profiles obtained by the PCAWG Working Group 11 (PCAWG-11) using a consensus approach combining six different state-of-the-art copy number calling methods[45]. GC content corrected LogR values were extracted from intermediate Battenberg algorithm results[46], smoothed using a running median, and transformed into copy number space according to $n = [2^{logR}(2(1 − \rho) + \psi\rho) − 2(1 − \rho)]/\rho$, where $\rho$ and $\psi$ are the PCAWG-11 consensus tumour purity and ploidy, respectively.

*Identification of telomeric deletions associated to HBV insertion events.* Single read clusters, identified with v-TraFiC, supporting an HBV insertion event (i.e., clusters of discordant read-pairs – Illumina – with apparently no reciprocal cluster within the proximal 500 bp, and whose mates support a somatic HBV event), were interrogated for the presence of associated telomeric deletions. Briefly, we looked for copy number loss calls from PCAWG (see "Copy number dataset" above) where: (i) the copy number loss extends from the HBV insertion breakpoint up to the end of the chromosomal arm, involving the telomere, and (ii) one independent cluster, which supports the integration of the HBV event, unequivocally demarcates the copy number loss boundary. We used MEIBA v0.8.8[15] to reconstruct the relevant insertion breakpoint.

*HBV rate across genomic features.* We analysed the frequency of HBV events across genomic features. $\chi^2$ was employed to test significance. For each test, we randomly generated in-silico HBV insertions in the genome, as many as observed in real dataset, with the regioneR package v1.22.0[47]. In cases where $\chi^2$ test assumptions were not met due to the low number of observations (i.e., cells with less than five observations), the $\chi^2$ test was not performed. False Discovery Rate (FDR) correction was applied. Chromatin segments: we used the ENCODE segmentation of the HepG2 cell line genome into a set of major genome states[48], considering the following two predicted categories: repressed and transcriptionally active; Replication timing: replication timing was defined using HepG2 Repli-seq data (GEO:GSM923446). Regions with gene expression signals higher than 70 were defined as early, below 20 were defined as late, and between 20 and 70 as middle, in the same way described by[20]; Expression: Transcripts Per Kilobase Million (TPM) from 208 liver tissues[49] were used to evaluate HBV insertion rate within genes classified according to different gene expression categories in the liver [0–0.5], [0.5–10], [10–10,000].

*Timing of viral insertions.* We have used two (related) ways of estimating the timing of HBV insertions and their mediated telomeric deletions. The first is a relative timing approach, which classifies insertion events as clonal early/late/NA or subclonal, depending on their allele frequency and the local copy number state. This approach does not look at the whole-genome doubling state of the tumour and only provides timing information relative to a chromosomal gain (if any). The second approach takes this one step further. When a whole-genome duplication has generated the gain of the HBV-derivative chromosome, we can time the gain itself much more precisely by aggregating info from the allele frequencies of small variants across the genome. By focusing on the clock-like mutations, this relative timing can be anchored and turned into a real-time timing.

We employed the SVclone algorithm v0.2.2[23] to obtain the number of reads supporting and non-supporting HBV DNA insertions. To deal with HBV insertions supported by single-read clusters only, a modification of the method was implemented to accept structural variants with only one break-end side as follows: (i) relevant filters were switched off in order to allow insertion events with one breakpoint only to be considered by SVclone; (ii) only two types of reads were extracted from the BAM file: split reads (soft-clipped reads that cross each break-end) and normal reads (reads that cross or span the break-ends but match the reference), being spanning reads removed (read pairs that align either side of the break-ends but match the reference). Read counts from SVclone, together with tumour purity and copy number states, were used as input of MutationTime.R v0.1[23] for the classification of HBV insertions into four different timing categories, namely clonal [early], clonal [late], clonal [NA] or subclonal. Then, real-time estimates for whole-genome duplication (WGD) events, based on CpG>TpG mutations analysis[23], were used to place particular HBV insertion within a chronological time-frame – in years – during a patient's lifespan, depending on whether mutations occurred before or after a WGD event.

*Probe synthesis and fluorescence in situ hybridisation.* Two sets of bacterial artificial chromosome (BAC) clones (RP5-1125N11 and RP11-891N16 for t(1;11); and RP11-125F4 and RP11-652N13 for t(8;17)) were obtained from the BACPAC Resources Center (https://bacpacresources.org/) to develop two-colour single-fusion FISH probes to detect chromosome translocations. *ARID1A* deletion probe was develop with RP5-696E2 and RP11-372B18 BAC clones, and Metasystems #D-5103-100-OG probe was used to study *TP53* gene deletion. RP5-1125N11, RP11-125F4, and RP5-696E2 BACs were labelled with Spectrum-Orange, and RP11-891N16, RP11-652N13 and RP11-372B18 with Spectrum-Green. FISH analyses were performed using the Histology FISH Accessory Kit (DAKO) following the manufacturer's instructions (PMID: 25798834 DOI: 10.1038/onc.2015.70) on 5 mm TMA sections mounted on positively charged slides (Thermo Scientific). Briefly, the slides were first deparaffined in xylene and rehydrated in a series of ethanol. Slides were pre-treated in 2-[N-morpholino]ethanesulphonic acid (MES), followed by a 30 min protein digestion performed on proteinase-K solution. After dehydration, the samples were denatured in the presence of the specific probe at 73 °C for 5 min and left overnight for hybridisation at 37 °C. Finally, the slides were washed with 20×SSC (saline-sodium citrate) buffer with detergent Tween-20 at 63 °C, and mounted on fluorescence mounting medium (DAPI in antifade solution). Cells were imaged with a Leica DM 5500B fluorescence microscope equipped with a 100x oil-immersion objective, Leica DM DAPI, Green and Orange fluorescence filter cubes and a CCD camera (Photometrics SenSys camera) connected to a PC running the Zytovision image analysis system (Applied Imaging Ltd., UK) with Z stack software v7.4. The z-stack images were manually scored by two independent investigators by counting the number of co-localised signals, representing fused transcripts, or missing signals, representing deletions, all over the tissue.

**Reporting summary**. Further information on research design is available in the Nature Research Reporting Summary linked to this article.

## Data availability

All genomic datasets generated for this manuscript were deposited in public databases. Somatic and germline variant calls, mutational signatures, subclonal reconstructions, transcript abundance, splice calls and other core data generated by the ICGC/TCGA Pan-cancer Analysis of Whole Genomes Consortium[13] are available at https://dcc.icgc.org/releases/PCAWG. Specifically, ONT sequencing bam files obtained in this study can be found in https://dcc.icgc.org/releases/PCAWG/pathogen_analysis. Additional information on accessing the data, including Illumina and ONT sequencing raw read files, can be found at https://docs.icgc.org/pcawg/data/. In accordance with the data access policies of the ICGC and TCGA projects, most molecular, clinical and specimen data are in an open tier, which does not require access approval. To access potentially identifying information, such as germline alleles and underlying sequencing data, researchers will need to apply to the TCGA Data Access Committee (DAC) via dbGaP for access to the TCGA portion of the dataset, and to the ICGC Data Access Compliance Office (DACO) for the ICGC portion. In addition, to access somatic SNVs derived from TCGA donors, researchers will also need to obtain dbGaP authorisation. The analyses in this paper used a number of datasets that were derived from the raw sequencing data and variant calls. The individual datasets are available at Synapse (https://www.synapse.org/), which are also mirrored at DCC portal (https://dcc.icgc.org). Donor clinical data, tumour histopathology, consensus CNA, consensus SV calls, driver mutational events, purity and ploidy, timed copy number segments and real time inferences of MRCA and WGD are available at Synapse with the following accession numbers: syn10389158, syn1038916, syn8042988, syn7596712, syn11639581, syn8272483, syn14778989 and syn14778990, respectively. Hyperlinks are show below. syn10389158, syn1038916, syn8042988, syn7596712, syn11639581, syn8272483, syn14778989, syn14778990.

## Code availability

A preliminary version of the code v-TraFiC for the identification of somatic HBV insertions, is available at http://gitlab.com/mobilegenomesgroup/v-TraFiC.

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

## Acknowledgements

We thank the Supercomputing Centre of Galicia (CESGA) for providing complementary computational resources. J.M.C.T is supported by European Research Council Grant ERC-2016-STG – ERC Starting Grant 716290, and Ministerio de Ciencia, Innovación y Universidades, Grants PGC2018-102245-B-100 and RYC-2014-14999. E.G.A. and P.O. are supported by Ministerio de Educacion Cultura y Deporte, fellowships FPU17/05396 and FPU18/03421, respectively. D.G-S. is supported by postdoctoral contract ED481B-2018/091 from Xunta de Galicia. B.R-M., A.O. and S.Z. are supported by Xunta de Galicia fellowships ED481A-2016/151, ED481A-2020/214 and ED481A-2018/199, respectively. A.L.B. is supported by Spanish Ministry of Economics and Competitiveness (MINECO) PhD fellowship BES-2016-078166. This work was supported by the Francis Crick Institute, which receives its core funding from Cancer Research UK (FC001202), the UK Medical Research Council (FC001202), and the Wellcome Trust (FC001202). For the purpose of Open Access, the authors have applied a CC BY public copyright licence to any Author Accepted Manuscript version arising from this submission. This project was enabled through access to the MRC eMedLab Medical Bioinformatics infrastructure, supported by the Medical Research Council (grant number MR/L016311/1). M.M-F. is funded by the Spanish Association for Cancer Research (AECC, grant207mart). S.R-P. is supported by grants from the Spanish National Research and Development Plan, Instituto de Salud Carlos III, and FEDER (PI17/02303, PI20/01837 and DTS19/00111); AEI/MICIU EXPLORA Project BIO2017-91272-EXP and AECC_Lab_2020 Project (Asociación Española Contra el Cáncer). X.F. and S.P.P. are supported by grants from the Instituto de Salud Carlos III through the Spanish National Plan for Scientific and Technical Research and Innovation (PI18/00079 and PI19/00036, respectively), co-funded by the European Regional Development Fund (ERDF), and from Secretaria d'Universitats i Recerca del Departament d'Economia i Coneixement (grant 2017_SGR_1753) and CERCA Programme/ Generalitat de Catalunya. J.R-B. is supported by a Rio Hortega Fellowship from the Institute of Health Carlos III (CM19/00087) and a 2020 TTD Research Grant from the Spanish Cooperative Group for the Teatment of Digestive Tumours (TTD). J.D. is a postdoctoral fellow of the Research Foundation – Flanders (FWO). U.G. is supported by Ministerio de Ciencia e Innovación grants PID2020-118970RB-100, SAF2016-75169-R and RYC-2014-15805. P.V.L. is a Winton Group Leader in recognition of the Winton Charitable Foundation's support towards the establishment of The Francis Crick Institute. A.D-L. is supported by a Juan Rodés contract from ISCIII (JR17/00016). A.R-C. is supported by GAIN and Xunta de Galicia (IN853B 2018/03).

## Author contributions

E.G.A., P.V.L., P.J.C. and J.M.C.T. conceived the study. E.G.A., J.D., Y.L., J.T., J.Z., C.J., A.B-O., B.R-M., K.H., S.Z., K.R., A.P.B., K.M., I.O., P.V.L. and J.M.C.T. contributed to pipelines and/or data management. E.G.A., J.D., J.Z., Y.L., C.J., M.T., K.H., P.V.L., P.J.C. and J.M.C.T. analysed the sequencing data. J.R-B., E.G.A., P.O., M.M-F. and J.M.C.T. analysed the clinical data. J.T., D.G-S., A.P-V., J.R-C, A.R-C. and A.D-L. performed sequencing. M.T., M.M-F., R.A., P.O., A.O. and A.L.B. performed laboratory experiments. M.T., H.N., X.F., S.P.P., A.O., H.A., K.C., M.U., S.H. and H.Y. provided tumour specimens and/or performed pathological diagnosis. D.G-S., U.G., C.R., M.G.B. and J.M.C.T. build a comprehensive biological model for HBV-mediated rearrangements. R.T. and S.R.P. performed cytogenetics. E.G.A., P.V.L., P.J.C. and J.M.C.T. wrote the manuscript with assistance from all authors. M.T., P.V.L., P.J.C. and J.M.C.T. supervised the project.

## Competing interests

The authors declare no competing interests.

## Additional information

[1]Centre for Research in Molecular Medicine and Chronic Diseases (CiMUS), Universidade de Santiago de Compostela, Santiago de Compostela 15706, Spain. [2]Department of Zoology, Genetics and Physical Anthropology, Universidade de Santiago de Compostela, Santiago de Compostela 15706, Spain. [3]The Francis Crick Institute, London NW1 1AT, UK. [4]Department of Human Genetics, University of Leuven, Leuven B-3000, Belgium. [5]The Biomedical Research Centre (CINBIO), Universidade de Vigo, Vigo 36310, Spain. [6]Department of Veterinary Medicine, University of Cambridge, Cambridge CB3 0ES, UK. [7]Cancer Epigenomics, Translational Medical Oncology Group (Oncomet), Health Research Institute of Santiago (IDIS), University Clinical Hospital of Santiago de Compostela (CHUS/SERGAS), Santiago de Compostela 15706, Spain. [8]Roche-Chus Joint Unit, Translational Medical Oncology Group (Oncomet), Health Research Institute of SantiagodeCompostela(IDIS), Santiago de Compostela 15706, Spain. [9]Centro de Investigación Biomédica en Red Cáncer (CIBERONC), Madrid 28029, Spain. [10]Wellcome Sanger Institute, Wellcome Genome Campus, Hinxton, Cambridge CB10 1SA, UK. [11]Department of Gastroenterology and Metabolism, Graduate School of Biomedical and Health Sciences, Hiroshima University, Hiroshima, Japan. [12]Collaborative Research Laboratory of Medical Innovation, Graduate School of Biomedical and Health Sciences, Hiroshima University, Hiroshima, Japan. [13]Research Center for Hepatology and Gastroenterology, Graduate School of Biomedical and Health Sciences, Hiroshima University, Hiroshima, Japan. [14]RIKEN Center for Integrative Medical Sciences, Yokohama, Kanagawa 230-0045, Japan. [15]Department of Surgery II, Wakayama Medical University, Wakayama, Japan. [16]Liver Unit, Hospital Clínic, University of Barcelona, IDIBAPS, CIBERehd, Barcelona, Spain. [17]Department of Molecular and Cellular Biology, Centro Nacional de Biotecnología – Consejo Superior de Investigaciones Científicas (CNB – CSIC), Madrid 28049, Spain. [18]Department of Medical Oncology, University Clinical Hospital of Santiago de Compostela, University of Santiago de Compostela, Santiago de Compostela 15706, Spain. [19]Translational Medical Oncology Group (Oncomet), Health Research Institute of Santiago de Compostela (IDIS), Santiago de Compostela 15706, Spain. [20]Molecular Cytogenetics and Genome Engineering Group, Centro Nacional de Investigaciones Oncológicas (CNIO), Madrid, Spain. [21]Division of Hematopoietic Innovative Therapies, Centro de Investigaciones Energéticas, Medioambientales y Tecnológicas (CIEMAT), Madrid, Spain. [22]Department of Haematology, University of Cambridge, Cambridge CB2 2XY, UK. [23]These authors contributed equally: Jonas Demeulemeester, Paula Otero, Clemency Jolly, Daniel García-Souto. [24]These authors contributed equally: Urtzi Garaigorta, Peter J. Campbell, Hidewaki Nakagawa, Peter Van Loo. ✉email: jose.mc.tubio@usc.es

