## [Peer review file · Nature Communications]

REVIEWER COMMENTS

Reviewer #1 (Remarks to the Author): Expert in cancer genomics, long read sequencing, and bioinformatics

In this manuscript, Álvarez et al. analyzed human hepatocellular carcinoma (HCC) genomes with HBV integration using both short read and long read sequencing data. They successfully detected canonical and non-canonical insertion patterns of HBV DNA. They also detected genomic changes including non-homologous chromosomal rearrangements and mega-base deletions which might be caused by HBV insertions. These events may occur in the earlier stage before whole genome duplication and parts of those events would be associated with driver events, such as TP53 and ARID1A disruption. I think that the authors should additionally show overall features of non-canonical insertions and impact of the observed events for HCC biology and clinical settings. There are some points that need to be addressed to improve the manuscript as below;

Points;

1. It is difficult to find information on the number for "8% of all HCC tumours" which is described in Abstract. Please clearly summarize the number of the cases analyzed by short or long read sequencers and these breakdown (i.e. insertion-negative, with canonical only, with non-canonical).

2. The authors showed canonical and non-canonical patterns of insertions at whole genomes in Fig. 1A. What genomic regions (genic/intergenic, repetitive, etc.) tend to be affected? This information is informative to characterize the non-canonical insertion events.

3. The authors should describe the reason why all insertions were not completely sequenced in Sup Fig. 2. I think that long read sequences can cover full-length HBV insertions. Did HBV insertions have any deletions?

4. The authors also mentioned that parts of the insertions resulted in megabase-size deletions including telomeric regions. Are there any differences between such "functional" insertions and the others? This information is important for elucidating mechanisms and biological relevance of the events. Please add description and discussion.

5. The authors pointed association between rearrangements by HBV integration and driver events. What percent of the rearrangement-positive cases could be associated with driver events, such as TP53 and ARID1A disruptions? This information can describe comprehensiveness and significance of the observed events.

6. The authors should describe clinical relevance of the mutational process described in this paper. Please describe differences between the cases focused in this study (8%?) and the others about histopathological features, pathological stage, prognosis and so on?

Minor points;

1. Please describe the meaning of "MCN" in Fig. 4A.

2. In Fig. 4B, the authors showed timing estimation of HBV insertions before HCC diagnosis. I think that information on diagnosis of hepatitis is also informative. Please discuss about association between timing of HBV infection and that of integration/WGD occurrence if possible.

Reviewer #2 (Remarks to the Author): Expert in hepatocellular carcinoma genomics and HBV

The authors show using hepatocellular carcinoma (HCC) samples from the Pan Cancer Analysis of

Whole Genomes project and both short and long-read next generation sequencing, that hepatitis B virus (HBV) integrations into host DNA lead to a number of major chromosomal changes, including non-homologous chromosomal fusions and megabase telomeric deletions. They provide evidence to suggest that the effects of HBV integration occur early in tumorigenesis and can be directly oncogenic. While similar events have been shown to occur in both cell lines and primary tumors, this manuscript clearly delineates the formation of interchromosomal rearrangements, including one that generates a dicentric chromosome. Studies using real-time estimation suggest that the insertion and rearrangement events occur many years before HCC diagnosis. Rearrangements are identified that are associated with loss of TP53 and ARID1A, which were confirmed by FISH.

Major concerns:

While the observations are well founded and elegantly described, novelty is somewhat limited by similar findings in recent publications, including Peneau et al., Gut 2021.

Reviewer #3 (Remarks to the Author): Expert in hepatocellular carcinoma

In the present manuscript, Alvarez et al. used multiplatform sequencing technologies to detect the genomic alterations induced by hepatitis B virus (HBV) in hepatocellular carcinoma (HCC). The authors identified some mutational mechanisms induced by HBV integration into the tumor genome. These alterations resulted in non-homologous chromosomal fusions and megabase-size telomeric deletions. These alterations were detected in at least 8% of HCC lesions. The authors conclude that HBV induces several genomic aberrations by its integration in the HCC genome. This event is active early during liver cancer evolution and might provide the driver rearrangements that a cancer clone needs to induce invasive HCC.

The study by Alvarez et al. is exciting and provides intriguing insights into the molecular pathogenesis of HBV-related HCC. The use of state-of-the-art technologies is undoubtedly a significant strength of the investigation. However, the study was not conducted in-depth, and no mechanistic studies were performed. Specifically, several different alterations were detected, generally rarely occurring, in the tumor genome. No studies in vitro and/or in vivo were conducted to demonstrate that the unraveled alterations are truly oncogenic and drive cancerogenesis. In other words, are these genomic aberrations induced by HBV integration driving events in hepatocarcinogenesis or passenger (irrelevant) alterations? Thus, the present study is purely descriptive in its current version and does not significantly extend our knowledge of the molecular mechanisms whereby HBV contributes to hepatocarcinogenesis.

Reviewer #1

In this manuscript, Álvarez et al. analyzed human hepatocellular carcinoma (HCC) genomes with HBV integration using both short read and long read sequencing data. They successfully detected canonical and non-canonical insertion patterns of HBV DNA. They also detected genomic changes including non-homologous chromosomal rearrangements and mega-base deletions which might be caused by HBV insertions. These events may occur in the earlier stage before whole genome duplication and parts of those events would be associated with driver events, such as TP53 and ARID1A disruption. I think that the authors should additionally show overall features of non-canonical insertions and impact of the observed events for HCC biology and clinical settings. There are some points that need to be addressed to improve the manuscript as below;

Points;

1. It is difficult to find information on the number for “8% of all HCC tumours” which is described in Abstract. Please clearly summarize the number of the cases analyzed by short or long read sequencers and these breakdown (i.e. insertion-negative, with canonical only, with non-canonical).

We may not have specified this clear enough in the main text. We have analysed a total of 296 HCC tumours from the PCAWG dataset, of which 17% (51/296) have at least one HBV integration event. The number of HCC samples with HBV-mediated telomeric deletions is 23 out of 296 (~8%). Nine samples were sequenced with ONT. We now include this information in main text (**page 8**), and a new **Supplementary table 1** details the number and type (canonical or non-canonical) of HBV integrations found per sample, and the sequencing technologies (short or long reads) employed for the analysis.

2. The authors showed canonical and non-canonical patterns of insertions at whole genomes in Fig. 1A. What genomic regions (genic/intergenic, repetitive, etc.) tend to be affected? This information is informative to characterize the non-canonical insertion events.

Three types of analysis were performed to answer this query. First, we have analysed the distribution of HBV insertions in 10 Mb windows across the HCC genome. This analysis revealed considerable variation in the rate of HBV integration. One region at chromosome 5, with 13 HBV insertions, stands out over the others, with a hotspot in the *TERT* gene (11 HBV insertions) (see new **Fig. 4a**). Second, we performed an analysis of HBV integration rate and genomic features, including replication timing, chromatin state and gene expression. Despite the low number of HBV insertions included in this analysis (n=148), we find a significant depletion of HBV events in intergenic regions and an enrichment in genes (**Fig. 4b**). Notably, 66% (93/148) of all HBV events fall in genes. Third, when we split HBV events into those linked to megabase-size telomeric deletions (n=40) and other HBV insertions (n=108), we observe that while the category of HBV insertions with no telomeric deletions exhibits the expected pattern for canonical HBV insertions [PMDI 29861854], characterized by an enrichment in regions of early replication timing, HBV insertions at telomeric deletions are depleted in those areas (**Fig. 4c**). This suggests different DNA repair mechanisms driving the integration of HBV DNA at different stages of the S phase. We now include a new section in the main text (**pages 9-10**) called “HBV insertion rate varies across the HCC genome”, together with a new **Fig. 4**, a new **Supplementary Fig. 5**, and a new section in the methods (**page 30**).

3. The authors should describe the reason why all insertions were not completely sequenced in Sup Fig. 2. I think that long read sequences can cover full-length HBV insertions. Did HBV insertions have any deletions?

We think there is a misunderstanding here, as those insertions were completely sequenced. In Supplementary Fig. 2, as in the remaining relevant figures, ONT reads were cut (discontinued lines) for illustrative purposes only. We now clarify this in figure legends (**Fig. 2, Fig. 6, SFig. 1, SFig. 2**) and include two files with the nucleotide sequences and coordinates of all the ONT reads represented in the figures; see **Supplementary Data 1** (sequences in FASTA) and **Supplementary Table 3** (sequences annotation).

4. The authors also mentioned that parts of the insertions resulted in megabase-size deletions including telomeric regions. Are there any differences between such “functional” insertions and the others? This information is important for elucidating mechanisms and biological relevance of the events. Please add description and discussion.

We have analysed the nucleotide sequence of every HBV insertion fully sequenced with long reads (n=22) and we observe that the sequences of HBV insertions mediating telomeric deletions tend to have a more complex structure than canonical insertions, including internal deletions, inversions and duplications of the reference sequence. However, based on this data, we cannot elucidate a molecular mechanism for the formation of these events. We now include new information in main text (**page 8-9**), and in a new **Supplementary Fig. 4**.

5. The authors pointed association between rearrangements by HBV integration and driver events. What percent of the rearrangement-positive cases could be associated with driver events, such as TP53 and ARID1A disruptions? This information can describe comprehensiveness and significance of the observed events.

We find 40 HBV-mediated rearrangements with telomeric deletions involving 23 HCC samples. These rearrangements result in the loss of 244 genes classified as tumour suppressor genes or loss-of-function genes according to Cancer Gene Census [PMID 30293088] or Intogen [PMID 32778778], respectively (new **Supplementary Table 4**). Thirty-seven of these genes are linked to hepatocellular carcinoma. Notably, at least three of the HBV-mediated telomeric deletions described in this manuscript are reported in the Compendium of driver somatic copy number alterations (SCNAs) published by the Pan-Cancer Analysis of Whole Genomes (PCAWG) consortium [see PMID 32025007]. This includes one event in sample SA501424 that removes *ARID1A* (**Fig. 7b**), an event in SA529830 that removes *TP53* (**Fig. 6**), and the one that removes *RBI* in SA501511 (**Fig. 3b** and **SFig. 2c**). In addition, we report four additional HBV-mediated telomeric deletions involving loss of three tumour suppressor genes that, although they are not reported in the PCAWG Compendium of SCNAs, they represent clonal events involving the loss of at least one relevant cancer-related gene in HCC, which includes *ARID1A* in SA501481 (**Fig. 7a**), *RPS6KA3* in SA529726 (**Fig. 3a**) and *IRF2* in SA268027 and SA269383 (**SFig. 3**). Although *IRF2* is not catalogued as a driver of HCC in neither CENSUS nor Intogen, functional studies have identified it as a cancer suppressor gene in HCC [PMDI 23264911]. Thus, the proportion of rearrangements described here that are associated with driver genes is 18% (7/40). These events are summarized in the table below. We now include this information in the main text (**page 13**).

Sample	Deletion coordinates	Missing cancer gene
SA501424	chr1:1-31503628	ARID1A
SA501481	chr1:1-57256948	ARID1A

SA529830	chr17:1-14933981	TP53
SA501511	chr13: 38591685-115169878	RB1
SA529726	chrX:1-150837858	RPS6KA3
SA268027	chr4:173136730- 191154276	IRF2
SA269383	chr4:172915474-191154276	IRF2

6. The authors should describe clinical relevance of the mutational process described in this paper. Please describe differences between the cases focused in this study (8%?) and the others about histopathological features, pathological stage, prognosis and so on?

We performed Kaplan-Meier survival analyses to evaluate the evolution of patients with HBV-mediated telomeric deletions versus other HCC patients with HBV insertions, and versus all the remaining HCC patients with and without HBV insertions (see figures below). The curves show no significant differences between the cohorts. It is most likely that other large deletions and rearrangements present in both cohorts could be influencing the results, making it difficult to evaluate the impact of this mutational mechanism in patients' evolution and disease prognosis. In addition, the analysis of the clinical data does not show clear evidence for association between the number of HBV-mediated telomeric deletions and tumour stage nor tumour grade. To better understand the clinical relevance of this mutational mechanism, we will need to survey the topography of somatic HBV insertions on a considerable larger scale, across thousands of HCC genomes together with high quality clinical data. However, we believe the clinical relevance of this study is in the analysis of the timing of HBV-mediated telomeric deletions (see next paragraph).

Figure. Kaplan-Meier survival curves. (Left) patients with HBV-mediated telomeric deletions (red), and patients with HBV insertions but without HBV telomeric deletions (blue). Y axis represents survival probability and X axis Time of survival in days. The table below shows the number of cases at risk along time (in days). **(Right)** patients with HBV-mediated telomeric deletions (red), and all remaining patients with HBV insertions and without HBV insertions but no HBV telomeric deletions (blue).

Chronic hepatitis B is a complex disease caused by HBV, which is characterized by the progressive degeneration of the liver tissue that eventually develops into HCC. The figure below, modified from PMID 28778687, defines four stages of disease progression with associated factors represented. Antiviral therapy is crucial for the clinical management of the disease. Current antiviral drugs, including nucleoside/nucleotide analogues (NAs) and interferon- α (INF- α) can suppress HBV replication and slow the progression of liver disease. Indicators to begin antiviral treatment are generally based on the levels of HBV DNA in serum, HBsAg (Hepatitis B surface antigen), levels of alanine transaminase (ALT, and enzyme released by dying hepatocytes) and the severity of liver disease, together with other factors such as age. For instance, in the immune tolerant stage of the disease in patients with no-fibrosis or minimal fibrosis under 30 years old,

current professional society guidelines recommend the initiation of antiviral therapies only when there is evidence of liver damage, which is defined by elevated levels of ALT above upper limits of normal. However, patients in this stage of the disease progression typically have very high levels of HBV DNA and HBsAg in serum, which means that the virus is replicating and HBV DNA insertions, including HBV-mediated telomeric deletions, are occurring. Our study reveals that HBV-mediated telomeric deletions are events that can drive the oncogenic process and occur early in HCC evolution, sometimes up to decades before cancer diagnosis. Thus, it is most likely that these early events that contribute to the initiation of the oncogenic process occur before antiviral therapies are applied, when HBV DNA levels are high and there is no apparent liver damage. The **clinical relevance** of our results is that they suggest that HBV infected patients in the immune tolerant stage could benefit from the administration of earlier antiviral therapies, which would suppress HBV replication, HBV integration and driver HBV events, including HBV-mediated telomeric deletions. We now include new information in the main text (discussion in **page 14**).

Figure. Phases of chronic HBV infection. The natural history of chronic hepatitis B infection is complex and can be divided into four phases defined by three clinical parameters: serum alanine aminotransferase (ALT) concentrations, serum HBV DNA levels, and Hepatitis B e antigen (HBeAg). The first phase, called “immune-tolerant” is characterized by the presence of HBeAg and high serum HBV DNA but normal ALT levels. The immune-tolerant phase is followed by the “HBeAg-positive immune-active” phase, when ALT levels become elevated above upper limits of normal. Seroconversion from HBeAg to hepatitis B antibody occurs, and a majority of patients transition to the “inactive carrier” phase, during which ALT levels return to normal and serum ALT DNA level are low or undetectable. In some patients, serum HBV DNA and ALT levels become elevated again, after years or decades. These patients are in the “HBeAg-negative immune-active” phase, characterised by fluctuating HBV DNA and ALT levels. Note that antiviral therapy is only recommended in chronic hepatitis phases when both viral load and transaminase levels are high. HBsAg denotes Hepatitis B surface antigen.

Minor points;

1. Please describe the meaning of “MCN” in Fig. 4A.

MCN means mutation copy number. We now note this in new **Fig. 5** (old Figure 4) legend.

2. In Fig. 4B, the authors showed timing estimation of HBV insertions before HCC diagnosis. I think that information on diagnosis of hepatitis is also informative. Please discuss about association between timing of HBV infection and that of integration/WGD occurrence if possible.

Time of HBV infection is not available. We do not know when infection have occurred along patient's lifespan, unfortunately.

Reviewer #2

The authors show using hepatocellular carcinoma (HCC) samples from the Pan Cancer Analysis of Whole Genomes project and both short and long-read next generation sequencing, that hepatitis B virus (HBV) integrations into host DNA lead to a number of major chromosomal changes, including non-homologous chromosomal fusions and megabase telomeric deletions. They provide evidence to suggest that the effects of HBV integration occur early in tumorigenesis and can be directly oncogenic. While similar events have been shown to occur in both cell lines and primary tumors, this manuscript clearly delineates the formation of interchromosomal rearrangements, including one the generates a dicentric chromosome. Studies using real-time estimation suggest that the insertion and rearrangement events occur many years before HCC diagnosis. Rearrangements are identified that are associated with loss of TP53 and ARID1A, which were confirmed by FISH.

Major concerns:

While the observations are well founded and elegantly described, novelty is somewhat limited by similar findings in recent publications, including Peneau et al., Gut 2021.

It is true that similar findings (i.e., rearrangements mediated by HBV) have been recently described by Peneau et al. However, we believe our work is still novel for the following reasons:

1. We find patterns on the Illumina paired-end sequencing data (namely, independent clusters of reads) that have not been described by Peneau et al. These patterns, which identify the boundaries of HBV-mediated rearrangements, are characterized in **Fig. 1c**, making them easier to find in future studies.
2. We use cutting-edge bioinformatic approaches to perform both relative and real-time timing estimation of the HBV-mediated rearrangements, finding that some rearrangements appear early in HCC evolution, sometimes decades before cancer diagnosis (**Fig. 5**). These timing methods are not employed by Peneau et al., which in our opinion makes a big difference.
3. We describe the formation of dicentric chromosomes as a consequence of this mechanism (**Fig. 3b**), which are unstable structures that in cancer can lead to breakage-fusion-bridge cycles and oncogene amplification (see, for instance, PMID: 32024998). These structures are not described by Peneau et al.
4. We find recurrent deletion of tumour suppressor gene *ARID1A* targeted by this mutational mechanism (**Fig. 7**). *ARID1A* represents one of the most important cancer genes that drive evolution of hepatocellular carcinoma. Mutations in *ARID1A* are not identified by Peneau et al. Moreover, in this new version of the manuscript, we report four additional HBV-mediated telomeric deletions involving loss of other three tumour suppressor genes in HCC, including *RBI*, *RPS6KA3* and *IRF2*. Note that although *IRF2* is not catalogued as a driver of HCC in the

Cancer Gene CENSUS nor Intogen, functional studies by the group of Dr Jessica Zucman-Rossi have identified it as a cancer suppressor gene in HCC [see PMDI 23264911]. We now include this information in main text (**page 13**).

Reviewer #3

In the present manuscript, Alvarez et al. used multiplatform sequencing technologies to detect the genomic alterations induced by hepatitis B virus (HBV) in hepatocellular carcinoma (HCC). The authors identified some mutational mechanisms induced by HBV integration into the tumor genome. These alterations resulted in non-homologous chromosomal fusions and megabase-size telomeric deletions. These alterations were detected in at least 8% of HCC lesions. The authors conclude that HBV induces several genomic aberrations by its integration in the HCC genome. This event is active early during liver cancer evolution and might provide the driver rearrangements that a cancer clone needs to induce invasive HCC.

The study by Alvarez et al. is exciting and provides intriguing insights into the molecular pathogenesis of HBV-related HCC. The use of state-of-the-art technologies is undoubtedly a significant strength of the investigation. However, the study was not conducted in-depth, and no mechanistic studies were performed. Specifically, several different alterations were detected, generally rarely occurring, in the tumor genome. No studies in vitro and/or in vivo were conducted to demonstrate that the unraveled alterations are truly oncogenic and drive cancerogenesis. In other words, are these genomic aberrations induced by HBV integration driving events in hepatocarcinogenesis or passenger (irrelevant) alterations? Thus, the present study is purely descriptive in its current version and does not significantly extend our knowledge of the molecular mechanisms whereby HBV contributes to hepatocarcinogenesis.

We share reviewer's interest in using in vitro and in vivo HBV infection model systems to confirm the oncogenic potential of the HBV-induced chromosomal rearrangements described in this study. Unfortunately, HBV is a very special virus with highly oncogenic potential in humans but with very limited capacities in vitro or in animal models, so it is not possible to use these models for those confirmatory experiments. On one hand, only a handful of hepatic tumoral cell lines (i.e., HepG2-NTCP and Huh-7-NTCP) and primary human hepatocytes (PHH) are susceptible to HBV infection, and even though HBV can integrate in these experimental systems (PMID 29437961) they are not suitable for oncogenic studies because either they are tumoral cell lines already or they de-differentiate and cannot be kept longer than 10-12 days in culture (in the case of PHH). On the other hand, naturally HBV can only infect humans and chimpanzees in vivo. For ethical reasons, research in chimpanzees is no longer possible and analysis of human samples has been the focus of the current manuscript. In the last decades, liver chimeric humanized mouse model systems have been developed. These models are susceptible to HBV infection (PMID 20179355; PMID 11283864). However, HBV does not induce hepatocellular carcinoma in these systems probably due to the lack of adaptive immune system in these models, making them not suitable for oncogenic studies.

Despite that we cannot employ in vitro or in vivo models to test the driver role of the mechanism reported in this manuscript, we believe there is already evidence of its oncogenic role. Relevant tumour suppressor losses described here were previously catalogued by the Pan-Cancer Analysis of Whole Genomes (PCAWG) consortium as driver events [PMDI 32025007]. Obviously, not all mutations in recurrently mutated cancer-related genes are drivers. For that reason, the PCWAG consortium created a Compendium of driver copy number alterations (SCNAs), as follows [methods in PMDI 32025007]. Briefly, GISTIC peaks found to be significant from the analysis of

cohorts of 23 tumour types analysed by TCGA were first retrieved. Then, regions of the relevant peaks were defined based on associating overlapping peaks across different cancers into metapeaks and, when possible, associating each metapeak with a driver gene that provides a consistent location at which to assess the DNA copy number. Finally, a metapeak was annotated as a driver and kept in the Compendium only if a significant shift in expression coherent with the type of SCNA, amplification or deletion, was observed. Using this Compendium, the PCAWG consortium identified a number of high confident driver copy number events in the PCAWG dataset [PMID 32025007]. These driver events include at least three deletions described in this manuscript, including the loss of *ARID1A* in sample SA501424 (**Fig. 7b**), the loss of *TP53* in sample SA529830 (**Fig. 6**), and a loss of one copy of *RBI* in SA501511 (**Fig. 3b** and **SFig. 2c**). Moreover, according to the data reported by the PCAWG, for two of these genes (*TP53* and *RBI*) an inactivating point mutation has been identified in the second allele of the gene (see, for instance, **SFig. 6**).

Although this Compendium has catalogued many driver SCNAs in the PCAWG dataset, it is obvious that some/many other regions remain missing. In this new version of the manuscript, we report four additional HBV-mediated telomeric deletions involving loss of three tumour suppressor genes that, although they are not reported in the PCAWG Compendium of SCNAs, they represent clonal events involving the loss of at least one relevant cancer-related gene in HCC, which includes *ARID1A* in SA501481 (**Fig. 7a**), *RPS6KA3* in SA529726 (**Fig. 3a**) and *IRF2* in SA268027 and SA269383 (**SFig. 3**). Although *IRF2* is not catalogued as a driver of HCC in the Cancer Gene CENSUS nor Intogen, functional studies have identified it as a cancer suppressor gene in HCC [PMID 23264911]. We now include this information in main text (**page 13**).

REVIEWERS' COMMENTS

Reviewer #1 (Remarks to the Author):

In the revised manuscript, the authors additionally characterized the HBV insertion events and their mediated telomeric deletions in detail and reported four additional important events in three tumor suppressor genes. They also advocated importance of identification of the "timing" of HBV-mediated deletions as clinical relevance and discussed that early administration of antiviral therapy is important for HCC prevention. I think that the manuscript become improved.

Minor:

I'm concerned that the timing of HBV-mediated deletions was just estimated by WGD timing analysis. I recommend that confirmation of the timing by different ways from WGD analysis should be conducted (or at least discussed).

Reviewer #3 (Remarks to the Author):

The authors have satisfactorily addressed the concerns raised by the reviewer.

September 24, 2021

In the revised manuscript, the authors additionally characterized the HBV insertion events and their mediated telomeric deletions in detail and reported four additional important events in three tumor suppressor genes. They also advocated importance of identification of the "timing" of HBV-mediated deletions as clinical relevance and discussed that early administration of antiviral therapy is important for HCC prevention. I think that the manuscript become improved.

Minor:

I'm concerned that the timing of HBV-mediated deletions was just estimated by WGD timing analysis. I recommend that confirmation of the timing by different ways from WGD analysis should be conducted (or at least discussed).

We thank the reviewer for the constructive criticism of our manuscript. We have in fact used two (related) ways of estimating the timing of HBV insertions and their mediated telomeric deletions. The first is our relative timing approach, which classifies insertion events as clonal early/late/NA or subclonal, depending on their allele frequency and the local copy number state (**Fig. 5a**). This approach does not look at the whole-genome doubling state of the tumour and only provides timing information relative to a chromosomal gain (if any). Our second approach takes this one step further. When a whole-genome duplication has generated the gain of the HBV-derivative chromosome, we can time the gain itself much more precisely by aggregating info from the allele frequencies of small variants across the genome. By focusing on the clock-like mutations, this relative timing can be anchored and turned into a real-time timing (**Fig. 5b**). Unfortunately, we are not aware of any other approaches to estimate the timing of HBV insertions and their mediated telomeric deletions from single tumour samples. Multi-region or time-series data could in theory provide further timing information if the loss has not occurred prior to the evolution of the most recent common ancestor. Our analyses indicate however, that these events typically occur several years prior to the initial diagnosis, and as such, time-series or multi-region samples are unlikely to increase resolution in the relevant timeframe. We now specify this better in the methods section of our manuscript (page 26, paragraph highlighted in yellow).